# Alteration of the Intra- and Inter-Lobe Connectivity of the Brain Structural Network in Normal Aging

**DOI:** 10.3390/e22080826

**Published:** 2020-07-28

**Authors:** Chi-Wen Jao, Jiann-Horng Yeh, Yu-Te Wu, Li-Ming Lien, Yuh-Feng Tsai, Kuang-En Chu, Chen-Yu Hsiao, Po-Shan Wang, Chi Ieong Lau

**Affiliations:** 1Institute of Biophotonics, National Yang-Ming University, Taipei 112, Taiwan; c3665810@ms24.hinet.net (C.-W.J.); ytwu@ym.edu.tw (Y.-T.W.); b8001071@yahoo.com.tw (P.-S.W.); 2Department of Research, Shin Kong Wu Ho-Su Memorial Hospital, Taipei 111, Taiwan; 3Department of Neurology, Shin Kong Wu Ho-Su Memorial Hospital, Taipei 111, Taiwan; M001074@ms.skh.org.tw (J.-H.Y.); M002177@ms.skh.org.tw (L.-M.L.); 4School of Medicine, College of Medicine, Fu Jen Catholic University, New Taipei 242, Taiwan; yuhfeng.tsai@gmail.com; 5Brain Research Center, National Yang-Ming University, Taipei 112, Taiwan; 6Department of Neurology, School of Medicine, College of Medicine, Taipei Medical University, Taipei 110, Taiwan; 7Department of Diagnostic Radiology, Shin Kong Wu Ho-Su Memorial Hospital, Taipei 112, Taiwan; leo_eraser@yahoo.com.tw; 8Department of Gastroenterology, Shin Kong Wo Ho-Su Memorial Hospital, Taipei 112, Taiwan; alkec55@gmail.com; 9Health Management Center, Shin Kong Wu Ho-Su Memorial Hospital, Taipei 111, Taiwan; 10Department of Neurology, Taipei Municipal Gan-Dau Hospital, Taipei 112, Taiwan; 11Dementia Center, Department of Neurology, Shin Kong Wu Ho-Su Memorial Hospital, Taipei 112, Taiwan; 12Applied Cognitive Neuroscience Group, Institute of Cognitive Neuroscience, University College London, London 01322, UK; 13University Hospital, Taipa 112, Macau

**Keywords:** aging, brain structural network, intra-lobe connectivity, inter-lobe connectivity

## Abstract

The morphological changes in cortical parcellated regions during aging and whether these atrophies may cause brain structural network intra- and inter-lobe connectivity alterations are subjects that have been minimally explored. In this study, a novel fractal dimension-based structural network was proposed to measure atrophy of 68 parcellated cortical regions. Alterations of structural network parameters, including intra- and inter-lobe connectivity, were detected in a middle-aged group (30–45 years old) and an elderly group (50–65 years old). The elderly group exhibited significant lateralized atrophy in the left hemisphere, and most of these fractal dimension atrophied regions were included in the regions of the “last-in, first-out” model. Globally, the elderly group had lower modularity values, smaller component size modules, and fewer bilateral association fibers. They had lower intra-lobe connectivity in the frontal and parietal lobes, but higher intra-lobe connectivity in the temporal and occipital lobes. Both groups exhibited similar inter-lobe connecting pattern. The elderly group revealed separations, sparser long association fibers, commissural fibers, and lateral inter-lobe connectivity lost effect, mainly in the right hemisphere. New wiring and reconfiguring modules may have occurred within the brain structural network to compensate for connectivity, decreasing and preventing functional loss in cerebral intra- and inter-lobe connectivity.

## 1. Introduction

As the human lifespan extends, aging is considered one of the greatest risk factors for neurodegenerative diseases, characterized by progressive loss of either the structure or function of neurons [1]. Morphologically, the structure of the brain changes in various ways as we age. Numerous morphological measures, including gray matter (GM) volume [2], cortical thickness [3], and surface area [4], have been applied to assess brain structural alterations during aging. Voxel-based morphometry (VBM) is one of the volumetric measures that has been widely used for assessment of normal aging, involving structural brain magnetic resonance imaging (MRI) [5]. However, the VBM method is a whole-brain technique that is essentially a qualitative rather than quantitative analysis, which may not be suitable for individual and local measurements [6]. Although differences in cortical thickness have been shown to be related to aging, this method was observed to present global sex differences and regionally greater cortical thickness in women, mostly in the left frontal, left parietal, and right parietal regions, as well as other small regions [7,8,9,10]. Moreover, the surface area method has also verified that inter-individual differences in surface area are strongly influenced by sex differences [11]. Hence, a sensitive morphological measure with less sex-related effect is crucial for detecting topological changes in healthy brain aging.

Complex natural structures, such as the cortical sulcus gyrus, can be difficult to quantify. The fractal dimension (FD), proposed by Benoit Mandelbrot in 1967 [12], is a quantitative parameter that characterizes the morphometric variability of a complex object [6]. An FD descriptor can offer quantitative information related to cortical convolution, while changes in the FD value reportedly indicate cortical abnormalities [6]. The FD analytic method is superior to conventional volumetric methods for quantifying the structural changes in white matter (WM) and GM because it exhibits smaller variances and less sex-related effect [6]. Hence, after the first proposal, FD has been extensively used to quantify shape complexity and morphological changes in cerebral MRI and has been widely used to describe structural complexity alterations in neurological or psychiatric disease [6,13,14,15]. The FD measure has regional variability that reflects local differences in brain structure. FD is complementary to volumetric measures and may assist in identifying disease state or disease progression [16]. Furthermore, FD has been shown to be a promising means for describing the morphology of cortical structures across different neurologic and psychiatric conditions, with good sensitivity in capturing atrophy processes [17].

In neuroscience, network analysis provides rich quantitative insights into the organization, development, and function of complex brain networks [18]. Detecting modules in a network may help identify relevant substructures that correspond to important functions, providing a link between structure and function in complex networks [19]. A cortical feature-based network, including the thickness [20], volume [21], and surface area [22], can highlight the anatomical connection between parcellated regions in neural degenerative diseases, such as dementia [23], multiple sclerosis [24], and Alzheimer’s disease [25]. Recently, we proposed a novel three-dimensional FD-based brain structural network to detect intra- and inter-modular connectivity alterations in brain lobes of spinocerebellar ataxia type 3 (SCA3) [26]. First, the FD can accurately assess the significantly atrophied, parcellated cortical regions. Second, the constructed network was revealed precisely, presenting the intra- and inter-modular connectivity of each brain lobe and the alterations between healthy individuals and patients with SCA3 [26]. In neurodegenerative disease or normal aging, monitoring brain structural network changes may be beneficial in understanding the pathophysiology and enabling the possibility of preventing or modifying progressive neurodegeneration.

Until now, with regard to healthy aging, the morphological changes in details of the parcellated cortical regions and whether these atrophies may cause brain structural network connectivity alteration have been minimally explored. In this study, we investigated and compared brain structural network connectivity between healthy groups of middle-aged individuals (30–45 years old) and in the early elderly age group (50–65 years old). First, the Desikan Region labels atlas was used to parcellate the cerebrum into 68 subregions as relative focal regions [27]. FD values for the parcellated regions of two groups were measured and compared with the assessment of the significant atrophied regions. We then calculated the covariance correlation of three-dimensional (3D) FD values between paired brain regions to produce a 68 × 68 correlation matrix for both groups. These correlation matrices were subsequently used in modular analysis to establish a structural connectivity network and compute the intra-modular and inter-modular connectivity of the whole brain and its lobes. Network properties, such as the modular numbers, modularity, and clustering coefficient of each group, were also measured and compared. We hypothesized that structural network alteration, reorganization, and intra- and inter-lobe connectivity changes may occur during aging.

## 2. Materials and Methods

### 2.1. Participants

A total of 100 healthy middle-aged adults (aged 30–45 y; 37.2 ± 4.9 y; male/female ratio, 50/50) and 100 healthy elderly adults (aged 50–65 y; 56.8 ± 3.7 y; male/female ratio, 50/50) participated in this study. All the participants were recruited from the Department of Neurology at Shin Kong Wu Ho Su Memorial Hospital from 2015 to 2019. No significant differences in age (*p* = 0.607) or sex were observed between the two groups. All participants were confirmed by neurologists that had no diseases of the central nervous system and no neurological abnormalities during the study period. The study protocol (code: 202000104R) was approved by the Institutional Review Board of Shin Kong Wu Ho-Su Memorial Hospital on 9 April 2020.

### 2.2. Image Acquisition and Cortical Feature–Based Structural Network

Brain axial MRI encompassing the entire cerebrum and cerebellum was performed using a 3-T Vision scanner (Philips Medical Systems, Amsterdam, The Netherlands). The participants were scanned using a circularly polarized head coil to obtain T1-weighted images (repetition time, 8.875 ms; echo time, 4.079 ms; matrix size, 384 × 384; 0.9-mm sagittal slices; field of view, 256 × 256 mm; voxel size, 0.67 × 0.67 × 0.9 mm^3^). Each structural MRI dataset was normalized to a presegmented and validated volumetric template by using DiffeoMap, and each normalized image volume was then segmented into GM, WM, and cerebrospinal fluid in the native space by using the SPM8 toolbox (Welcome Center for Human Neuroimaging, UCL Queen Square Institute of Neurology). The cortex was then parcellated and aligned into 68 distinct regions of interest (ROIs) with the Desikan–Killiany cortical atlas (DK atlas) [27] structures by using the FreeSurfer toolbox in MATLAB R2019b software (MathWorks, Natick, MA, USA). A total of 34 cerebral subregions (ROIs) were present in each hemisphere in the DK atlas. We then rearranged the ROIs of the DK atlas according to the lobes to which they belong and placed them in the order of the frontal lobe (1–28, odd numbers for left hemisphere and even numbers for right hemisphere), temporal lobe (29–46), parietal lobe (47–60), and occipital lobe (61–68). Table 1 summarizes the details of ROIs and abbreviations of the rearranged DK atlas.

### 2.3. FD Analysis and Brain Structural Network

FD has been extensively applied to investigate morphological changes in the cerebral cortex caused by neurological diseases [16,17,18]. The procedure of FD computation is as follows. Let *N*(*r*) denotes the minimal number of cubes of size *r* covering the fractal object. The power law relationship that defines the FD of a fractal is given by $N\left(r\right)\alpha \;r^{-FD}$ [6,28,29]. This implies that a larger *N*(*r*) or FD should cover a more irregular fractal object. The relation can be expressed as a line equation, log(N(r))=FD∗log(1r)+k, meaning that the value of FD can be estimated from the slope of the line. There are three steps in the FD estimation procedure. In the first step, cubic boxes of size *r* (edge length in pixel size) are selected and stacked side by side to encompass the whole 3D fractal object. The count is set to 1 whenever a box encompasses any pixel belonging to the fractal object and 0 otherwise. In the second step, the total number of nonempty boxes *N*(*r*) required to completely cover the whole fractal object is calculated. Then, *r* is progressively decreased and the same counting process is repeated. In the final step, the estimated FD value serves as the slope value of the regression line between log(*N*(*r*)) and log(1/*r*). Because FD analysis is based on a logarithmic scale, a more complex cerebral structure will have a higher FD value, whereas a degeneration of the cerebral structure may reveal a decrease in the cerebral FD value [6]. Hence, the FD can facilitate the evaluation of the tiny morphological changes in the brain cortex in normal aging. There are two steps to establish the brain structural network. First, the covariance correlation between paired brain regions is calculated to indicate the strength of interregional connections. The calculated 68 × 68 covariance correlation matrix of 3D FD values for paired regions is then applied to build the brain structural network. Second, the different brain regions are categorized into several modules based on their interregional connections using modular analysis [30].

### 2.4. Modular Analysis

In modular analysis, brain regions are categorized into several modules, where the connections are stronger within each module and weaker between modules. We defined the 68 anatomical brain regions as nodes and interregional covariance correlation coefficients as the edges between the nodes. The suitability of a modular partition can be measured using modularity, *Q*:(1)$$Q=\frac{1}{2m}\sum_{i,j}\left[A_{i,j}-\frac{k_{i}k_{j}}{2m}\right]\delta \left(c_{i},c_{j}\right),$$
where *A* is the connection matrix of the network and each element of *A* is the covariance correlation coefficient between regions; $k_{i}=\sum_{j}A_{ij}$ is defined as the sum of the covariance correlation coefficient between node *i* and its connected regions, and is also called the degree of node *I*; $m=\frac{1}{2}\sum_{i,\;j}A_{ij}$ represents the total number of edges; and *c_i_* denotes the module of node *i*. The *δ*-function *δ*(*i, j*) is 1 when nodes *i* and *j* belong to the same module and 0 otherwise. Here, *Q* represents the edge numbers of all paired nodes belonging to the same module. A larger *Q* implies a superior partition that is more likely to construct a modular organization [31].

In this study, we set a 20% proportion of the strongest correlation coefficients as a threshold, whereby all entries below the threshold, negative correlations, and all entries on the main diagonal (self-to-self connections) were set to 0, and the links did not exist. The participation coefficient and intra-modular degree for each region were defined as the indices of their inter- and intra-modular connection density, respectively [32].

#### 2.4.1. Intra-Modular Connectivity Analysis

Intra-modular connectivity is measured using the normalized within-module degree:(2)$$z_{i}=\frac{k_{i}-\bar{k_{ci}}}{\sigma_{k_{ci}}},$$
where $k_{i}$ is the number of edges linking the *i*th node to other nodes in its module *c*; $\bar{k_{ci}}$ is the average of $k_{i}$ of all nodes in module *c*; and $\sigma_{k_{ci}}$ is the SD of the intra-modular degrees of all nodes in module *c*. Thus, a higher value *z_i_* represents a stronger intra-modular connectivity for node *i*. In addition, nodes with high intra-modular degrees are considered hubs, whereas the other nodes are considered nonhubs. The intra-modular connectivity for the whole network (*Z_total_*) is the mean *Z_i_* of all 68 nodes. Similarly, the intra-modular connectivity for each lobe, namely *Z_frontal_*, *Z_parietal_*, *Z_temporal_*, or *Z_occipital_*, represents the average of *Z_i_* within the respective lobe.

#### 2.4.2. Inter-Modular Connectivity Analysis

Inter-modular connectivity is measured using the participation coefficient:(3)$$\;P_{i}=1-\sum_{c=1}^{C}{\left(\frac{k_{ci}}{k_{i}}\right)}^{2},$$
where $k_{ci}$ is the number of edges connecting the *i*th node to other nodes in its module *c*, and $k_{i}$ is the number of degrees in node *i* in the network. The participation coefficient ranges from 0 to 1. A participation coefficient of 0 represents a largely intra-modular connection within node *i*, whereas that close to 1 indicates a largely inter-modular connection of node *i*. The inter-modular connectivity for the whole network (*P_total_*) is the mean *P_i_* of all 68 nodes. Similarly, the inter-modular connectivity for each lobe, namely *P_frontal_*, *P_parietal_*, *P_temporal_,* or *P_occipital_*, represents the average of *P_i_* within the respective lobe.

### 2.5. Statistical Analysis

In this study, a two-tailed *t* test and multiple false discovery rates (FDR) correction were used to determine whether significant differences existed between study groups for 3D FD values and network parameters. Note that each group with 100 subjects had 100 3D FD values for each region. We computed the 3D FD value on the basis of correlation between any two regions. As a result, a 68 × 68 correlation map was obtained for each group to build a structural network, resulting in one set of topological properties for each structural network. Accordingly, we could not directly perform any statistical comparison of the corresponding topological properties between these two structural networks. In this study, a permutation test was conducted to statistically compare the differences in network properties between the two groups [33]. To test the null hypothesis, we randomly selected 25 women and 25 men from each study group and reassigned these 50 subjects as the randomized middle-aged group and randomized elderly group, separately. This randomized simulation and recalculation of the network properties were repeated 1000 times to recomputed the correlation matrix for each randomized group. The 95th percentile points of each distribution of the 1000 simulations were used as critical values in a two-sample, one-tailed *t* test to reject the null hypothesis, with a type I error probability of 0.05. Then, the network properties *Q*, *P*, and *Z* were calculated for each reassigned correlation matrix of the two groups. Following the permutation process, 1000 sets of network parameters were used in a two-sample, one-tailed *t* test with FDR correction to assess significant differences between the study groups.

## 3. Results

### 3.1. Healthy Elderly Group Exhibited More Significant Atrophy in the Left Brain Hemisphere

Table 2 summarizes the measured FD values of each lobe and the ROIs of significantly different FD values between the two groups. The elderly group exhibited significant atrophy in the bilateral frontal, temporal, and parietal lobes, but increased FD in the occipital lobe. The right parietal lobe was the most significantly atrophied lobe (*p* = 0.00025). The elderly group exhibited more significant atrophy in the frontal and temporal lobes in the left hemisphere (frontal lobe *p* values [L/R], 0.0336/0.0374; temporal lobe *p* values [L/R], 0.003/0.034). In parcellated regions, the elderly group exhibited 14 significantly atrophied ROIs in the frontal, temporal, and parietal lobes. Significant bilateral cerebral atrophy was observed in the elderly group; these atrophied ROIs were mainly in the left hemisphere (left/right, 9/5). The elderly group also had a significantly increased FD value in the bilateral pericalcarine cortices (PerCa) of the occipital lobe. Figure 1 illustrates the locations of ROIs of significantly different FD values between the middle-aged and elderly groups.

### 3.2. Elderly Group Exhibited Lower Modularity Values, Less Dense Modules, and More Separated Networks

Figure 2 illustrates the correlation map of the rearranged Desikan–Killiany atlas between different lobes for the middle-aged (Figure 2a) and elderly groups (Figure 2b). The color bar indicates the intensity of the correlation, ranging from 0 (blue) to 1 (red). The subregions (ROIs) of frontal lobe regions are illustrated from 1 to 28 on the *x*-axis and *y*-axis (odd numbers denote ROIs in the left hemisphere and even numbers denote ROIs in the right hemisphere) and marked within the red squares. The temporal lobes are within the purple squares (*x*-axis, 29:46; *y*-axis, 29:46), the parietal lobes are within the blue squares (*x*-axis, 47:60; *y*-axis, 47:60), and the occipital lobes are within the green squares (*x*-axis, 61:68; *y*-axis, 61:68). Clearly, the elderly group has a sparse, less dense correlation map. The correlation ratios of each lobe in the elderly group are smaller than those in the middle-aged group. Compared with the middle-aged group, the elderly group reveals much sparser links (lower correlation ratio) between lobes, which may imply functional dissociation between lobes in the elderly group. 

The brain modular networks for the middle-aged and elderly groups are summarized in Table 3 and Table 4, respectively. Both groups have five modules in the brain structural network. The brain structural network of the middle-aged group contained larger module (module 1, 22 nodes), three medium-sized modules (module 2, 14 nodes; module 3, 13 nodes; and module 4, 11 nodes), and one small module (module 5, 8 nodes). The elderly group exhibited network patterns differing from those of the middle-aged group. Their brain connecting links were less dense and could be grouped into one larger module (module 1, 19 nodes), two medium-sized modules (module 2 and module 3, 17 nodes), and two smaller modules (module 4, 8 nodes; module 5, 7 nodes). For each group, module 1 was the largest and the dominant module of the brain network.

### 3.3. Modular Structures of the Middle-Aged Adult Brain Network

The brain structural network in the middle-aged group has five functionally oriented modules. Module 1 in the middle-aged group contains 14 frontal nodes, 3 temporal nodes, 3 parietal nodes, and 2 occipital nodes. Module 1 contained four bilateral nodes (highlighted with underlining in Table 3) of the frontal lobe (CACg(L,R), CMF(L,R), Tr(L,R) and PreC(L,R)) which mainly correspond to regions associated with cognitive and executive functions [26,34]. Module 2 comprises 14 regions, including the frontal, temporal, parietal, and occipital lobes, which are mainly related to sensorimotor, visual, and spatial functions. Module 3 comprises 13 nodes in the frontal, temporal, and parietal lobes, including parahippocampal (PaH), superior temporal (SM), and supramarginal (SM) regions, which may be related to mnemonic and emotional functions [26]. We found that modules 1, 2, and 4 have at least one pair of bilateral node links, and the network modularity (*Q*) value for the middle-aged group was 0.2307.

The elderly group exhibited network patterns different from those of the middle-aged group. Their brain connecting links were less dense, although after analysis of 1000 permutations, the parcellated cerebral regions of the elderly group could be grouped into five network modules, similarly to the middle-aged group. A reorganized structural network occurred in the elderly group. The elderly group revealed segregation in the module component size and a lower modularity value (*Q*) of 0.2234 (*p* < 0.01). The lower modularity value suggested that its networks are comparatively sparse and inefficient. Module 1 contains 19 nodes (ROIs), including 8 frontal nodes, 3 temporal nodes, 5 parietal nodes, and 3 occipital nodes. The frontal and parietal lobes are the major component nodes in module 1, and these nodes are functionally related to cognitive and executive functions. Module 2 comprises six frontal nodes, six temporal nodes, four parietal nodes, and one occipital node. We found that module 2 contains major temporal lobe nodes, which are associated with auditory, language, and mnemonic functions. The six frontal nodes in module 2 include PreC(L), PreC(R), and CACg(R), which are associated with cognitive and executive functions.

After permutation analysis, we found re-modularization of network modules in the elderly group. The elderly group showed decreased component size of the modules in their brain structural network. One larger module (19 nodes), two medium-sized modules (17 nodes), and two smaller modules (8 nodes and 7 nodes) were grouped in the brain structural network of the elderly group. The nodes contained in each module were reorganized and less overlapped than the ones contained in the module of the middle-aged group. For instance, only four frontal nodes (Or(L), RoACg(R), Ins(L), and CMF(R)) in module 1 of the middle-aged group were included in module 1 of the elderly group. Nodes in module 1 of the middle-aged group were remodularized and separated into modules 1, 2, and 3 in the elderly group. Seven nodes (CMF(L),PreC(L) PreC(R),CACg(R),Tr(R), MT(L), PoCg(R), and PerCa(L)) in module 1 of the middle-aged group were grouped to module 2 of the elderly group. Another six nodes (CACg(L), LOrF(L), PaC(L), Tr(L), MT(L), and Lg(L)) in module 1 of the middle-aged group were grouped into module 3 of the elderly group. We further found only two modules (modules 1 and 2) in the elderly group to have at least one pair of bilateral nodes. The network modularity (*Q*) for the middle-aged group was 0.2307, whereas it was lower in the elderly group (0.2234; *p* < 0.01), suggesting that the networks in the elderly group were comparatively less dense and efficient.

Figure 3 and Figure 4 demonstrate the distribution of nodes in each module in the middle-aged group (Figure 3) and the elderly group (Figure 4) by using BrainNet Viewer software [35]. In each subfigure, module 1 is labeled with red dots, module 2 with yellow dots, module 3 with green dots, module 4 with turquoise dots, and module 5 with royal blue dots. In the left and right upper subfigures of the two groups, the red dots (module 1) of the middle-aged group are scattered in a larger area and reveal a denser connection between each node than in the elderly group. By contrast, module 2 (yellow dots) and module 3 (green dots) in the elderly group reveal a wider distribution and denser connection than module 2 and module 3 in the middle-aged group.

### 3.4. Elderly Group Revealed Significantly Decreased Intra-Modular Connectivity in Frontal and Parietal Lobes and Increased Intra-Modular Connectivity in Temporal and Occipital lobes

Figure 5 presents the intra-modular connectivity of each lobe in the middle-aged group (Figure 5a–d) and the elderly group (Figure 5e–h). In this study, we termed the intra-modular connectivity of each lobe as the intra-lobe connectivity. In each figure, the left half depicts the ROIs of each lobe in the left hemisphere, while the right half depicts those of each lobe in the right hemisphere. The blue lines depict the short association fibers of the left cerebral hemisphere, while the green lines depict the short association fibers of the right cerebral hemisphere. The commissural fibers (transverse fibers) between bilateral hemispheres are depicted by red lines. The width of the connecting line indicates the strength of the connectivity coefficient; a wider line implies a higher connecting strength between nodes. Table 5 summarizes the intra-lobe connectivity of the total lobe and each hemisphere lobe for the two groups.

First, both groups showed similar connecting patterns in all lobes. The middle-aged group demonstrated a lateralized effect of intra-lobe connectivity. They showed significantly higher intra-lobe connectivity in the left frontal and parietal lobes, but significantly lower intra-lobe connectivity in the left temporal and occipital lobes. Compared with the middle-aged group, the elderly group revealed fewer short association fibers bilaterally in the frontal and parietal lobes. The elderly group also revealed decreased commissural fibers in their frontal and parietal lobes. The elderly group had significantly lower intra-lobe connectivity in the frontal and parietal lobes and increased intra-lobe connectivity in the temporal and occipital lobes. In the temporal lobe, the elderly group had a higher intra-lobe connectivity of 0.4239, whereas the middle-aged group had a lower intra-lobe connectivity ratio of 0.3351. Comparing Figure 5b with Figure 5f, we found that the temporal lobe in the elderly group exhibited many more links of commissural fibers (B, En, Fu, IT, PaH, Pol, and TrT) than did those in the middle-aged group (B, ST, TrT); hence, increased intra-lobe connectivity in the temporal lobe may be anticipated in the elderly group. In the parietal lobe, as Figure 5g illustrates, the elderly group had fewer short association and commissural fibers and a lower intra-lobe connectivity ratio of 0.3378, whereas the middle-aged group had a higher intra-lobe connectivity of 0.3779 (*p* < 0.05) in the parietal lobe. Similar to the results related to temporal lobes, the elderly group exhibited increased intra-lobe connectivity in their occipital lobe, with a connectivity of 0.4642, whereas the middle-aged group had a lower connectivity of 0.3876 (*p* < 0.05). Similarly, the occipital lobe of the elderly group revealed denser links of short association and commissural fibers (PerCa, Lg) than the middle-aged group (PreCa).

### 3.5. Elderly Group Revealed Significantly Decreased Inter-Lobe Connectivity in All Lobes

The measured inter-lobe connectivity of all lobes and each hemisphere in the two groups is summarized in Table 6. The elderly group revealed significantly decreased inter-lobe connectivity in all lobes, especially in the parietal lobe of the right hemisphere (86.1%; *p* < 0.01). 

Figure 6 illustrates the inter-modular connectivity between lobes of the middle-aged group and the elderly group. Figure 6a,c illustrate the long association fiber of each lobe, while Figure 6b,d illustrate the commissural fibers (transverse fibers) crossing bilateral hemispheres. In each figure, the left half depicts the left hemisphere, while the right half depicts the right hemisphere. The hemispheres are further broken down into the frontal lobe, temporal lobe, parietal lobe, and occipital lobe. Each ROI of the frontal lobe is labeled with blue circles and abbreviations, the temporal lobes are indicated by green circles, red circles denote the parietal lobes, and occipital lobes are illustrated by green-blue circles. The width of the connecting line in each figure indicates the connectivity coefficient strength between nodes, and a wider line implies a higher connectivity between nodes. The blue lines indicate the uncinate and arcuate fibers (frontal lobe to temporal lobe)—frontoparietal connections are illustrated by chartreuse green lines, while the superior longitudinal fasciculus and occipitofrontal fasciculus (frontal lobe to occipital lobe and occipital lobe to frontal lobe) are illustrated by red lines. The yellow lines indicate the inferior longitudinal fasciculus fibers (occipital lobe to temporal lobe)—the temporal lobe and parietal lobe connecting fibers are indicated by moss green lines, while the purple lines indicate the connecting fibers between the occipital lobe and the parietal lobe.

A comparison of Figure 6a and Figure 6c shows that the elderly group had a connecting pattern almost similar to that of the middle-aged group, but it had thinner and fewer connecting lines between nodes. We found that the temporal lobe to occipital lobe (yellow lines in Figure 6c) and parietal lobe to occipital lobe (purple lines in Figure 6c) showed the highest decrease in connecting fibers during aging, including in long association fibers of lateral hemispheres and commissural fibers of bilateral hemispheres. The association fiber of the frontal lobe to parietal lobe in the right hemisphere (green line of right hemisphere in Figure 6c) also demonstrated the most decreased connectivity. These results may imply that the elderly group had a lateral long association connectivity loss effect, mainly in the right hemisphere. Interestingly, the elderly group had fewer long association and commissural fibers from the temporal lobe to the occipital lobe but more connectivity fibers from the temporal lobe to the parietal lobe in bilateral hemispheres, especially in the right hemisphere. The ratio of inter-lobe connectivity of the elderly group to the middle-aged group for the frontal lobe was 89.12%; for the temporal lobe, it was 92.3%; for the parietal lobe, it was 88.1%; and for the occipital lobe, it was 88.8%. In the elderly group, the temporal lobe had the least aging effect on inter-lobe connectivity alteration.

## 4. Discussion

In this study, we proposed an FD value–based structural network to investigate morphological changes or alterations of brain structural networks during healthy aging. We recruited 50 healthy male and 50 healthy female participants aged from 30 to 45 years as the middle-aged group and 50 healthy male and 50 healthy female participants aged from 50 to 65 years as the elderly group. The main findings of this study were as follows. Subjects in the elderly group exhibited significant lateralized atrophy in the left frontal lobe, temporal lobe, and parietal lobe. Most of these FD atrophied regions were included in the mainly transmodal regions of brain development and degeneration (“last-in, first-out”) model [36]. The elderly group exhibited lower modularity values, smaller component size modules, and fewer bilateral link networks. Regarding intra-lobe connectivity, the two groups demonstrated similar connecting patterns in every lobe, but in the elderly group, the intra-lobe connectivity was higher in the temporal lobe and the occipital lobe and lower in the frontal lobe and the parietal lobe. The middle-aged group exhibited lateralized intra-lobe connectivity symptoms and significantly higher connectivity in the left frontal and parietal lobes but lower connectivity in the left temporal and occipital lobes. Regarding inter-lobe connectivity, separation of inter-lobe connectivity was observed in the elderly group. They showed a connecting pattern similar to that of the middle-aged group, but lower connecting intensity, sparser linking lines, and decreased inter-lobe connectivity in all lobes.

### 4.1. Fractal Dimension-Based Brain Networks Can Manifest Morphological and Functional Modifications in Normal Aging

The human brain undergoes both morphological and functional modifications across the human lifespan. Understanding the aspects of brain reorganization that are critical in normal aging is crucial [37]. Brain network analysis, either functional or structural, can provide insights into associations and help detect structural or functional connectivity alterations between subregions. Meier et al. proposed a significant high goodness-of-fit level of brain network structure–function mapping and modality dependence between the connectivity matrices of the resting-state functional and structural networks [38]. Multiple statistically robust pattern analyses also reflected reliable combinations of structural and functional subnetworks that are optimally associated with one another [39]. These patterns generally do not show a one-to-one correspondence between structural and functional edges, but have many functional relationships arising from nonoverlapping sets of anatomical connections. The network organization of the cerebral cortex supports the emergence of diverse functional network configurations that often diverge from the underlying anatomical substrate. A structural network can predict a functional network, but the two networks do not necessarily overlap [39].

In the present study, we measured an FD value of 68 parcellated cerebral cortex ROIs in the middle-aged and elderly groups and constructed the brain structural networks. First, we found that the elderly group exhibited a tendency of left lateralized atrophy and that the left temporal lobe had the most atrophied ROIs. These results were compatible with previous normal aging studies [40,41]. During normal aging, the global WM and GM structure revealed that the maturity of the brain increases from youth to middle age, and then a degeneration process starts and continues through the elderly age range [41]. The maturity and degeneration symptoms revealed a quadratic pattern of change in FD values for the whole brain as well as for each hemisphere. The FD values for each hemisphere increased to a maximal value at the age of 50 years and then began to decrease thereafter, especially for the left hemisphere, which exhibited a more significant decreasing tendency than the right hemisphere [40]. In neuroscience, one model, termed “last in, first out,” posits, for instance, that healthy age-related brain degeneration mirrors development, with the areas of the brain thought to develop later also degenerating earlier [36,42]. This model defined a spatially specific network of mainly transmodal regions encompassing the heteromodal cortex and limbic and paralimbic regions—the lateral prefrontal cortex, frontal eye field, intraparietal sulcus, superior temporal sulcus, posterior cingulate cortex, and medial temporal lobe [42]. In this model, the onset age of the “first-out” stage is approximately 50 years, the same as the onset age of the elderly group in this study. We found that the FD atrophied regions in this study are included in the mainly transmodal regions of the “last-in, first-out” model. Therefore, our results of the FD measurement of brain degeneration may evidence and reflect the process of the “last-in, first-out” model.

After permutation analysis, we found that both groups had the same five modules in their brain structural networks. The middle-aged group revealed four larger modules (each module had more than 10 ROIs) and one smaller module of structural network, whereas the elderly group had a decreased component of module size for three larger and two smaller structural network modules. Reorganization and regrouping of the modules of the brain structural network occurred in the elderly group. Our results showed that the elderly group had modular organization and decreased network modularity with aging, which was in line with previous functional network studies [37,43]. The middle-aged group had more connecting lines between bilateral nodes in their modules than the elderly group did, which may provide the middle-aged group a better functional integration of the brain network than the elderly group. During aging, nodes in module 1 of the middle-aged group were redistributed into modules 1, 2, and 3 of the elderly group, and these processes occurred in other modules from the middle-aged group to the elderly group. We demonstrated that aging leads to a progressive and increasing reconfiguration of modules and redistribution across hemispheres. Finally, we identified the brain regions that most contribute to network reconfiguration and those that remain more stable across the lifespan.

### 4.2. Compensation Mechanism of Intra-Lobe Connectivity of Temporal and Occipital Lobes in Normal Aging

In the present study, the elderly group exhibited increased intra-lobe connectivity in the occipital lobe and decreased intra-lobe connectivity in the frontal and parietal lobes. The intra-lobe connectivity is the sum of all connectional weights within the lobe. A higher value of intra-lobe connectivity indicates greater significance of the corresponding module within the lobe. Globally, the elderly group had a lower modularity value than the middle-aged group (elderly group, 0.2234; middle-aged group, 0.2307). In aging, brain morphology plays a crucial role in the functional and structural reorganization [44]. Morphological changes in the brain may alter its network, and thus different local morphological changes result in a discrepancy of local connectivity, such as intra-lobe connectivity. The age-related increases in the structural connectivity of the occipital lobe have been reported in a previous cortical thickness network study [44]. Recently, a GM volume-based structural network study documented that only the temporal-lobe-related network showed a significant quadratic tendency of connectivity with age. The connectivity of the temporal-lobe-related network increased to a maximum value at 50 years and then decreased with age, whereas other trajectories of corresponding structural networks demonstrated significant linear decline tendencies with age [45]. Another GM structural network study also indicated that age was not significantly associated with GM volume in five networks—the temporal network, auditory network, and three cerebellar networks [46]. All of these studies suggested that the cortical morphology in the medial temporal lobe mediates the age effects on its structural connectivity.

Our findings may provide insights into these tendencies of age-related changes in structural networks. We found that the elderly group had significant atrophy in their temporal lobe. Moreover, in Figure 5f, which depicts the intra-lobe connectivity analyses of the temporal lobe, the elderly group exhibited not only more short association fibers but also more commissural fibers (transverse fibers) between nodes. Puxeddu et al. also reported observations similar to ours. They reported that brain areas, which reconfigure most across the lifespan, are located mainly on the lateral part of the cortex, specifically in temporal regions, motor and sensory areas, the parietal lobe, and the posterior cingulate cortex. The participation coefficient of the temporal lobe is positively correlated with age, so that nodes in the temporal lobe become more integrated in the network with age. The opposite happens for the parietal cortex, whose participation coefficient is negatively correlated with age and becomes more segregated with age [47]. The function of the network interaction is based on the structural connectivity; numerous studies have reported the relevance of short-range fibers in cognitive efficiency and brain activation in aging [48]. The short-range fibers are less myelinated, and thus are vulnerable to aging effects, whereas the long-range fibers have thicker myelination, which better insulates and protects the neuron and axon, thereby enhancing the resistance to aging effects [48]. Another diffusion tensor image (DTI) study compared age-related changes in association, commissural, and projection WM fiber regions, and reported that association fibers showed the most pronounced declines over time [49].

### 4.3. New Wiring and Reconfiguring Modules Occurred within the Brain Structural Network during Normal Aging

In this study, we measured the GM of 68 parcellated ROIs and verified that the association fibers of intra- or inter-lobe connectivity revealed more age-related decreases than commissural fibers. The correlation between microstructural WM properties (DTI) and macrostructural GM (volume) across normal aging was verified, with WM showing a strong correlation with GM along the aging process [50]. Our findings are in line with previously reported DTI studies. We further precisely detected the alteration of intra- or inter-lobe connectivity in different lobes and found that the temporal lobe and occipital lobe increased commissural fiber connectivity to prevent a short association connectivity decrease during aging processes. Numerous brain network studies have reported that temporal- and occipital-lobe-related networks revealed decreased aging effects on connectivity alteration, but the compensation mechanics of the intra-lobe connectivity of the brain structural network in the temporal lobe and occipital lobe have been less investigated. Our finding of an increase in the commissural fibers (transverse fibers) in the intra-lobe connectivity in the temporal lobe may reconfigure and enforce the intra-lobe structural network connectivity. This may reveal a compensatory mechanism in the elderly group to prevent aging effects on intra-lobe connectivity and functional loss in the temporal lobe and occipital lobe.

We also measured the inter-lobe connectivity between lobes, where the connectivity is the sum of connectional weights between lobes. A higher value of inter-lobe connectivity suggests a stronger connection between the two corresponding lobes. In each lobe, the elderly group had a lower inter-lobe connectivity, and the inter-lobe connectivity in the right hemisphere was lower than that in the left hemisphere. The long association fibers between frontal and parietal lobes in the right hemisphere exhibited the most decreased connectivity, although the elderly group showed decreased connectivity in long association fibers from the temporal lobe to the occipital lobe, but increased connectivity to parietal lobes. Among four cerebral lobes in the elderly group, we found that the temporal lobe revealed the latest aging effect of inter-lobe connectivity alteration. The human brain demonstrated three developmental waves of myelination during aging [51]. Most regions reached stability when individuals were in their 30s, and some regions reached stability when individuals were in their 50s. Age at onset of decline was also bimodal—in some right hemisphere regions, the curve declined after the age of 60, but in other left hemisphere regions no significant decline from the stable plateau was noted [51]. DTI analysis revealed that mean diffusivity is the most age-sensitive measure, with the strongest negative age associations in the association fibers. During aging, the WM microstructure across the brain tracts becomes increasingly correlated with older age and may reflect an age-related aggregation of systemic detrimental effects [52]. Furthermore, age-related WM deficits increased gradually from posterior to anterior segments within specific fiber tracts traversing the frontal and parietal cortices but not the temporal cortex [53]. Our results were consistent with those of these studies and revealed that lateral inter-lobe connectivity lost effect in the elderly group, mainly in the right hemisphere, although myelination of cerebral fibers occurred during aging and caused an associated fiber connectivity decrease. We found that during aging, new wiring and reconfiguring modules may have occurred within the brain structural network to compensate for the connectivity decrease and prevent functional loss in cerebral intra- and inter-lobe connectivity.

One recent concept termed as ‘‘brain reserve’’ has been proposed that some individuals may have increased ‘‘baseline neuroplasticity’’ to offer better circuit remodeling in response to various insults [54]. The concept of brain reserve may refer to the ability to tolerate the age-related changes and the disease-related pathology in the brain without developing clear clinical symptoms or signs [55]. Epidemiological studies revealed that high education, adult-life occupational work complexity, and intellectual challenge experiences may increase the individual ability of brain reserve [55]. This mentally and socially integrated lifestyle in late life is crucial for the occurrence of dementia symptoms and may postpone the onset of clinical dementia and Alzheimer’s disease (AD) [55]. Although our study is an inter-group comparison investigation, the compensation mechanisms of new wiring and reconfiguring modules of our results may reveal associations with the concept of ‘‘brain reserve’’ and imply that some individuals of the elderly group may have increased their “baseline adaptive neuroplasticity” and may exhibit “brain reverse” in normal aging.

## 5. Conclusions

In summary, the elderly group had a lower modularity value, smaller modules, and fewer bilateral association fibers in their brain structural network. They had lower intra-lobe connectivity in the frontal lobe and parietal lobe but higher intra-lobe connectivity in the temporal and occipital lobes. They exhibited an inter-lobe connecting pattern similar to that of the middle-aged group. In every lobe, the elderly group revealed separations, sparser long association fibers, and commissural fibers of inter-lobe connectivity. The elderly group revealed that lateral inter-lobe connectivity lost effect, mainly in the right hemisphere. New wiring and reconfiguring modules may have occurred within the brain structural network to compensate for connectivity decrease and to prevent functional loss in cerebral intra- and inter-lobe connectivity.

## Figures and Tables

**Figure 1 entropy-22-00826-f001:**
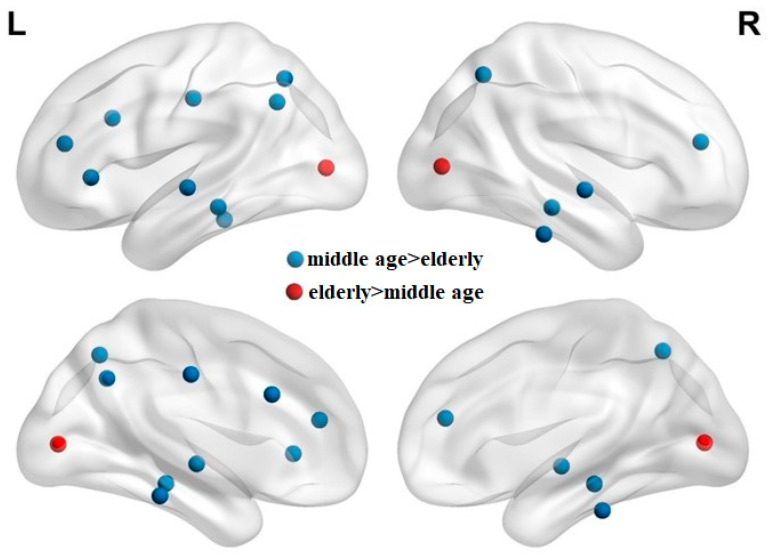
Locations of regions of interest of significantly different fractal dimension values between the middle-aged and elderly groups. The blue dots represent the significantly atrophied regions of interest (ROIs) in the elderly group, while the red dots represent the significantly increased ROIs in the elderly group.

**Figure 2 entropy-22-00826-f002:**
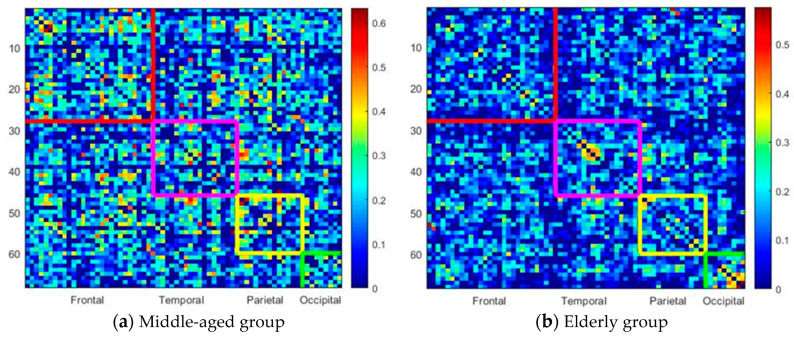
Correlation map of the rearranged Desikan–Killiany atlas between different lobes in the middle-aged group and the elderly group: (**a**) middle-aged group; (**b**) elderly group.

**Figure 3 entropy-22-00826-f003:**
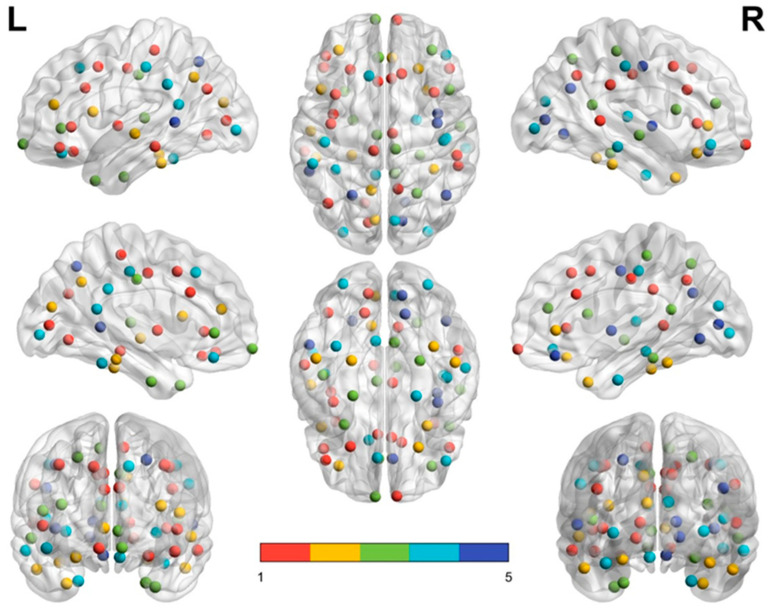
Middle-aged group modules. Module 1, 22 nodes in red; module 2, 14 nodes in yellow; module 3, 13 nodes in green; module 4, 11 nodes in turquoise; module 5, 8 nodes in blue.

**Figure 4 entropy-22-00826-f004:**
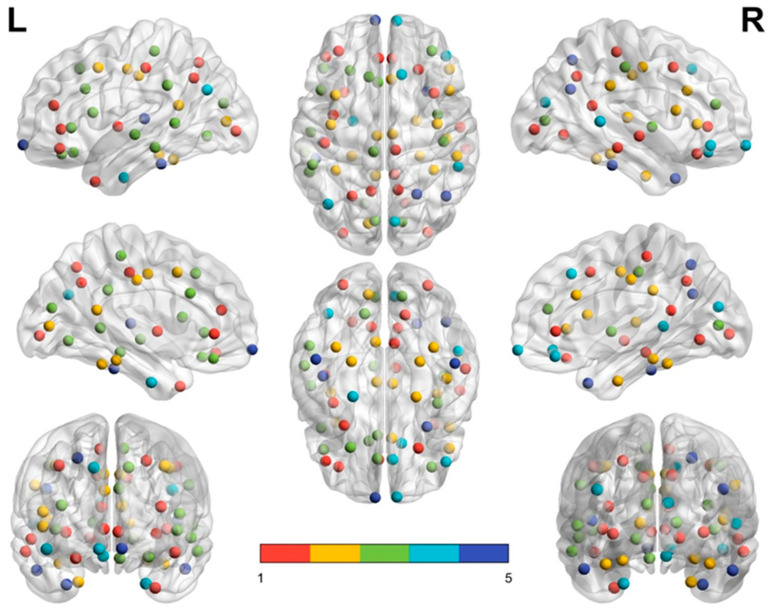
Elderly group modules. Module 1, 19 nodes in red; module 2, 17 nodes in yellow; module 3, 17 nodes in green; module 4, 8 nodes in turquoise; module 5, 7 nodes in blue.

**Figure 5 entropy-22-00826-f005:**
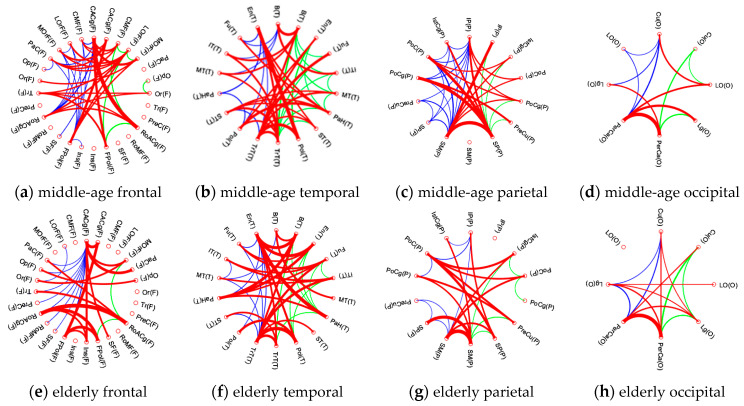
Intra-modular connectivity of each lobe in the middle-aged group. (**a**–**d**) Middle-aged group: (**a**) frontal lobe; (**b**) temporal lobe; (**c**) parietal lobe; (**d**) occipital lobe. (**e**–**h**) Elderly group: (**e**) frontal lobe; (**f**) temporal lobe; (**g**) parietal lobe; (**h**) occipital lobe.

**Figure 6 entropy-22-00826-f006:**
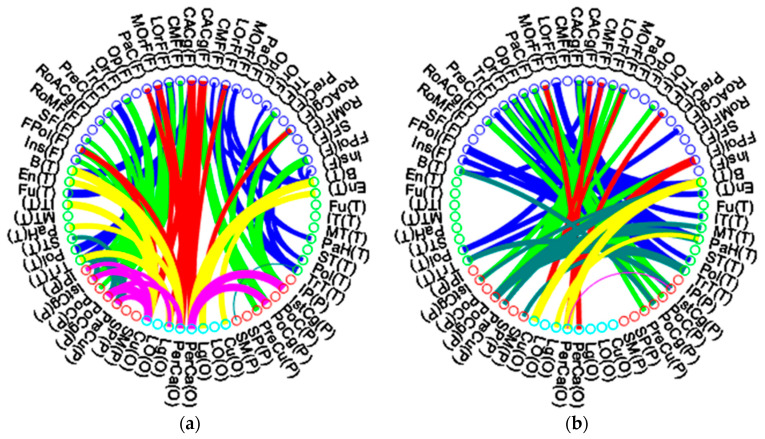
Inter-modular connectivity between lobes of the middle-age group and the elderly group: (**a**,**c**) the long association fiber of each lobe; (**b**,**d**) the commissural fibers (transverse fibers) crossing bilateral hemispheres.

**Table 1 entropy-22-00826-t001:** Regions of interest (ROIs) and abbreviations of the rearranged Desikan–Killiany atlas. ROIs of the frontal lobe (1–28, odd numbers denote the ROIs in the left hemisphere and even numbers denote the ROIs in the right hemisphere), temporal lobe (29–46), parietal lobe (47–60), and occipital lobe (61–68) are shown.

Frontal	ROI	Abbreviation	Temporal	ROI	Abbreviation
1, 2	Caudal anterior cingulate	CACg	37, 38	Middle temporal	MT
3, 4	Caudal middle frontal	CMF	39, 40	Parahippocampal	PaH
5, 6	Lateral orbito frontal	LOrF	41, 42	Superior temporal	ST
7, 8	Medial orbito frontal	MOrF	43, 44	Temporal pole	TPol
9, 10	Paracentral	PaC	45, 46	Transverse temporal	TrT
11, 12	Parsopercularis	Op	Parietal		
13, 14	Parsorbitalis	Or	47, 48	Inferior parietal	IP
15, 16	Parstriangularis	Tr	49, 50	Isthmus cingulate	IstCg
17, 18	Precentral	PreC	51, 52	Postcentral	PoC
19, 20	Rostral anterior cingulate	RoACg	53, 54	Posterior cingulate	PoCg
21, 22	Rostral middle frontal	RoMF	55, 56	Precuneus	PreCu
23, 24	Superior frontal	SF	57, 58	Superior parietal	SP
25, 26	Frontalpole	FPol	59, 60	Supra marginal	SM
27, 28	Insula	Ins	Occipital		
Temporal			61, 62	Cuneus	Cu
29, 30	Bankssts	B	63, 64	Lateral occipital	LO
31, 32	Entorhinal	En	65, 66	Lingual	Lg
33, 34	Fusiform	Fu	67, 68	Pericalcarine	PerCa
35, 36	Inferior temporal	IT			

**Table 2 entropy-22-00826-t002:** Measured FD (fractal dimension) values of each lobe of hemispheres for middle-aged and elderly groups and significantly different parcellated regions of elderly group.

FD Measure of Lobes between Middle-Aged and Elderly Groups				Significantly Different Parcellated Regions			
Lobes (L/R)	Middle-Age	Elderly	*p* Value	Regions	Middle-Age	Elderly	*p* Value
Frontal (F)	2.2399 ± 0.1035	2.2326 ± 0.1032	0.0221	F_CACg(L)	2.1081	2.0668	0.037
Frontal (L)	2.2375 ± 0.1091	2.2294 ± 0.1109	0.0336	F_Tr(L)	2.2499	2.2334	0.044
Frontal (R)	2.2423 ± 0.1017	2.2358 ± 0.0989	0.0374	F_RoMF(L)	2.4128	2.3997	0.005
Temporal (T)	2.1991 ± 0.1232	2.1897 ± 0.1244	0.0084	F_RoMF(R)	2.4128	2.4014	0.048
Temporal (L)	2.2065 ± 0.1145	2.1968 ± 0.1171	0.003	T_MT(L)	2.3118	2.2989	0.043
Temporal (R)	2.1918 ± 0.1379	2.1825 ± 0.138	0.034	T_MT(R)	2.3254	2.3103	0.018
Parietal (P)	2.2972 ± 0.1047	2.2858 ± 0.1078	0.0005	T_ST(L)	2.3267	2.3148	0.033
Parietal (L)	2.2953 ± 0.1055	2.2833 ± 0.1109	0.0026	T_ST(R)	2.3392	2.329	0.037
Parietal (R)	2.2990 ± 0.1122	2.2882 ± 0.1135	0.00025	T_PaH(L)	2.0762	2.0536	0.035
Occipital (O)	2.2033 ± 0.1306	2.2134 ± 0.1139	0.0267	T_IT(R)	2.3278	2.3138	0.031
Occipital (L)	2.1908 ± 0.151	2.2061 ± 0.1283	0.007	P_PoCg(L)	2.1961	2.1684	0.038
Occipital (R)	2.2159 ± 0.1287	2.2208 ± 0.1170	0.348	P_PreCu(L)	2.3302	2.3171	0.045
				P_SP(L)	2.3567	2.3487	0.031
				P_SP(R)	2.3567	2.3487	0.016
				O_PerCa(L)	2.0416	2.0935	0.0009
				O_PerCa(R)	2.0836	2.1097	0.04

**Table 3 entropy-22-00826-t003:** Modules of the brain structural network in the middle-aged group.

Middle Age Group	Lobe (Parcellated Regions (Left, Right))
Module 1 (22)	**Frontal:**CACg(L), CACg(R), CMF(L), CMF(R), LOrF(L), PaC(L), Or(L), Tr(L), Tr(R), PreC(L), PreC(R), RoACg(R), FPol(R), Ins(L) **Temporal:** MT(L), B(R), TPol(R) **Parietal:** IP(L), PoCg(R), SP(R) **Occipital:** Lg(L), PerCa(L)
Module 2 (14)	**Frontal:** MOrF(L), SF(L), Or(R), **Temporal:** Fu(L), En(R), MT(R), TrT(R) **Parietal:** PoC(L), SM(L), IstCg(L), PoC(R) **Occipital:** LO(L), LO(R), Cu(R)
Module 3 (13)	**Frontal:** RoACg(L), FPol(L), PaC(R), Op(R), SF(R), **Temporal:** En(L), TPol(L), TrT(L) PaH(R), ST(R) **Parietal:** PoCg(L), IstCg(R), SM(R)
Module 4 (11)	**Frontal:** Op(L), RoMF(L), RoMF(R), LOrF(R), **Temporal:** IT(L), PaH(L), ST(L), Fu(R), IT(R), **Parietal:** PreCu(L), **Occipital:** Cu(L)
Module 5 (8)	**Frontal:** MOrF(R), Ins(R), **Temporal:** B(L), **Parietal:** SP(L), IP(R), PreCu(R) **Occipital:** Lg(R), PerCa(R)

**Table 4 entropy-22-00826-t004:** Modules of the brain structural network in the elderly group.

Elderly Group	Lobe (Parcellated Regions (Left, Right))
Module 1 (19)	**Frontal:** Or(L), RoACg(L), RoACg(R), RoMF(L), Ins(L), CMF(R) LOrF(R) PaC(R) **Temporal:** TPol(L), MT(R), ST(R) **Parietal:** PoC(L), PreCu(L), SP(L), IstCg(R) preCu(R) **Occipital:** LO(L), LO(R), Lg(R)
Module 2 (17)	**Frontal:** CMF(L) PreC(L) PreC(R), CACg(R) OP(R) Tr(R) **Temporal:** Fu(L), PaH(L) En(R) Fu(R) PaH(R) TrT(R) **Parietal:** IstCg(L), PoCg(L), PoCg(R) SM(R), **Occipital:** PerCa(L)
Module 3 (17)	**Frontal:** CACg(L) LOrF(L), MOrF(L) PaC(L) Op(L) Tr(L) SF(L) RoMF(R) Ins(R) **Temporal:** B(L) MT(L) ST(L) **Parietal:** SM(L), PoC(R) **Occipital:** Cu(L), Lg(L), PerCa(R)
Module 4 (8)	**Frontal:** MOrF(R), Or(R), SF(R), FPol(R) **Temporal:** En(L), B(R), **Parietal:** IP(L), **Occipital:** Cu(R)
Module 5 (7)	**Frontal:** FPol(L) **Temporal:** IT(L) IT(R) TrT(L) TPol(R), **Parietal:** SP(R), IP(R)

**Table 5 entropy-22-00826-t005:** Intra-lobe connectivity of lobes and each hemisphere lobe for two groups.

Group	Frontal Lobe			Temporal Lobe			Parietal Lobe			Occipital Lobe		
	Total	Left	Right	Total	Left	Right	Total	Left	Right	Total	Left	Right
Middle age (M)	0.3986	0.4552	0.3421	0.3351	0.3127	0.3575	0.3779	0.426	0.3298	0.3876	0.3551	0.4207
Elderly (E)	0.3686	0.4168	0.3211	0.4239	0.3867	0.4488	0.3378	0.3576	0.3195	0.4642	0.4309	0.5012
Ratio (E/M)	92.4%	91.6%	93.8%	126.5%	123.7%	125.6%	89.4%	83.9%	96.9%	119.8%	121.3%	119.1%

**Table 6 entropy-22-00826-t006:** Inter-lobe connectivity in the lobes in each hemisphere between the groups.

Group	Frontal Lobe			Temporal Lobe			Parietal Lobe			Occipital Lobe		
	Total	Left	Right	Total	Left	Right	Total	Left	Right	Total	Left	Right
Middle age (M)	0.7119	0.7131	0.7106	0.7088	0.7087	0.7089	0.7104	0.7099	0.7108	0.7134	0.7137	0.7131
Elderly (E)	0.6345	0.636	0.633	0.6546	0.658	0.6512	0.6265	0.6405	0.6125	0.6341	0.6383	0.630
Ratio (E/M)	89.1%	89.2%	89.1%	92.3%	92.8%	91.9%	88.1%	90.2%	86.1%	88.8%	89.4%	88.4%

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
