# Peer review of "Alteration of the Intra- and Inter-Lobe Connectivity of the Brain Structural Network in Normal Aging"

_entropy, 2020, doi:10.3390/e22080826_

Round 1

Reviewer 1 Report

The paper is scientifically well written. The information is of interest and the authors allow a full understanding and interpretation of the data. I think that this paper can be accepted for publication

Author Response

We all authors thank and appreciate reviewer's valuable comment and suggestions. 

Reviewer 2 Report

Aging is considered as one of the greatest risk factors for neurodegenerative diseases characterized by progressive loss of either structure or function of the neurons. In this study, the authors have undergone a novel fractal dimension-based structural network was proposed to measure atrophy of parcellated cortical regions, and alteration of structural network parameters including intra- and inter-lobe connectivity were detected in a middle-aged group (30–45 y old) and an elderly group (50–65 y old). The fractal dimension is s superior to conventional volumetric methods as quantitative parameter that characterizes the morphometric variability of a complex object. The results suggest that the elderly group had a lower modularity value, smaller component size modules, and fewer bilateral association fibers in their brain structural network. They had lower intra-lobe connectivity in the frontal lobe and parietal lobe, but higher intra-lobe connectivity in the temporal and occipital lobes. They exhibited an inter-lobe connecting pattern similar to that of the middle-aged group. This is an interesting study. I suggest that authors shall discuss the impact of individual brain reserve on the results (see, DOI: 10.1007/S00702-013-1154-2).

Author Response

We thank reviewer's suggestion.

in the revised manuscript, we have added a paragraph to introduce the " brain reverse" issue and explain the association between this issue and our results in the discussion section. The added paragraph is as follow:

In recent years, the concept of ‘‘brain reserve’’ has emerged to describe some individuals having an increased ‘‘baseline adaptive neuroplasticity’’, providing greater dynamic capacity for adjusting and remodeling cortical circuits to various stressors [54]. The concept of brain reserve may refer to the ability to tolerate the age-related changes and the disease related pathology in the brain without developing clear clinical symptoms or signs [55]. Epidemiological studies revealed that high education, adult-life occupational work complexity, and intellectual challenges experiences may increase the brain reserve [55]. This mentally and socially integrated lifestyle in late life is crucial for the occurrence of dementia symptoms and may postpone the onset of clinical dementia and Alzheimer's disease (AD) [55]. Although our study is a between groups comparison investigation. The compensation mechanisms of new wiring and reconfiguring modules of our results may reveal associations with the concept of ‘‘brain reserve’’, and imply that some individuals of the elderly group may have increased their “baseline adaptive neuroplasticity” and exhibit “brain reverse” in normal aging.

We also have added two studies (1:DOI: 10.1007/S00702-013-1154-2) Freret, T.; Gaudreau, P.; Schumann-Bard, P.; Billard, J. M.; Popa-Wagner, A. Mechanisms underlying the neuroprotective effect of brain reserve against late life depression. 2015, Journal of Neural Transmission, 122(1), 55-61 (reviewer 2 suggested) and (2) Fratiglioni, L.; Wang, H. X. Brain reserve hypothesis in dementia. 2007Journal of Alzheimer's disease12(1), 11-22 to evidence our revised manuscript. We do believe these suggestions and revision will improve the quality of this article.
